# Identification of the Kinase-Substrate Recognition Interface between MYPT1 and Rho-Kinase

**DOI:** 10.3390/biom12020159

**Published:** 2022-01-18

**Authors:** Mutsuki Amano, Yoko Kanazawa, Kei Kozawa, Kozo Kaibuchi

**Affiliations:** 1Department of Cell Pharmacology, Graduate School of Medicine, Nagoya University, 65 Tsurumai, Showa-ku, Nagoya 466-8550, Aichi, Japan; kozawa.kei@med.nagoya-u.ac.jp (K.K.); kaibuchi@fujita-hu.ac.jp (K.K.); 2Institute for Comprehensive Medical Science, Fujita Health University, Toyoake 470-1192, Aichi, Japan; yoko.kanazawa@fujita-hu.ac.jp

**Keywords:** Rho-kinase, MYPT1, phosphorylation, substrate recognition, interface, docking motif

## Abstract

Protein kinases exert physiological functions through phosphorylating their specific substrates; however, the mode of kinase–substrate recognition is not fully understood. Rho-kinase is a Ser/Thr protein kinase that regulates cytoskeletal reorganization through phosphorylating myosin light chain (MLC) and the myosin phosphatase targeting subunit 1 (MYPT1) of MLC phosphatase (MLCP) and is involved in various diseases, due to its aberrant cellular contraction, morphology, and movement. Despite the importance of the prediction and identification of substrates and phosphorylation sites, understanding of the precise regularity in phosphorylation preference of Rho-kinase remains far from satisfactory. Here we analyzed the Rho-kinase–MYPT1 interaction, to understand the mode of Rho-kinase substrate recognition and found that the three short regions of MYPT1 close to phosphorylation sites (referred to as docking motifs (DMs); DM1 (DLQEAEKTIGRS), DM2 (KSQPKSIRERRRPR), and DM3 (RKARSRQAR)) are important for interactions with Rho-kinase. The phosphorylation levels of MYPT1 without DMs were reduced, and the effects were limited to the neighboring phosphorylation sites. We further demonstrated that the combination of pseudosubstrate (PS) and DM of MYPT1 (PS1 + DM3 and PS2 + DM2) serves as a potent inhibitor of Rho-kinase. The present information will be useful in identifying new substrates and developing selective Rho-kinase inhibitors.

## 1. Introduction

Eukaryotic protein kinases are a large family of highly conserved enzymes mediating the phosphorylation of substrate proteins in Ser/Thr/Tyr residues and they play key roles in the cellular signal transduction pathway. The activity of each protein kinase is temporally and spatially regulated, and aberrant activation or inactivation of kinase sometimes leads to disease. The catalytic domain of kinase specifically forms a complex with the target substrate and ATP. In addition to the catalytic domain, intra- and/or intermolecular domains relating to protein–protein interactions also contribute to the specification of substrates. Most protein kinases recognize the ‘consensus sequence’ surrounding the substrate phosphorylation site through the active site [1,2]. The consensus sequence is necessary, but not sufficient, for determining phosphorylation sites. For example, PKA preferably phosphorylates the ‘[R/K][R/K]x[S/T]Φ’ (Φ: hydrophobic residue) sequence and mitogen-activated protein kinase (MAPK) phosphorylates the ‘[S/T]P’ sequence, but the sequences satisfying the ‘consensus’ are not always phosphorylated by these kinases, suggesting that an additional rule is required [3]. Several kinases, including the MAPK family, interact with substrates through both the active site and the docking site within the catalytic domain, which enhances the specificity and efficiency of the phosphorylation of target proteins. At least two types of docking motifs that interact with the docking site have been found, from the many substrates that interact with MAPK family kinases, and have been shown to bind to the different docking sites of the kinase catalytic region [4,5]. Synthetic peptides derived from the sequences involved in the kinase–substrate interface, including pseudosubstrates, docking sites of kinases, and docking motifs of substrates have been developed as specific kinase inhibitors [6].

We have previously developed a novel method (named KISS analysis) of efficiently screening protein kinase substrates [7,8,9]. In this study, we found numerous candidate phosphorylation sites for nine protein kinases. Sequence alignment of candidate phosphopeptides demonstrated that the catalytic subunit of PKA preferably phosphorylates the [R/K][R/K]x[S/T]Φ sequence, as reported. The catalytic fragments of Rho-kinase and PKN also show almost the same sequence preference as that of PKA, but the overlapping phosphorylation sites among these three kinases were very limited [7], indicating that these kinases recognize their own substrates differently within the catalytic domain. This prompted us to identify novel interfaces between kinases and substrates, other than the active site and phosphorylation site. For this purpose, we focused on the interaction of Rho-kinase with its substrate MYPT1.

Rho-associated kinases (ROCKs) ROCK1 and ROCK2/Rho-kinase (we refer to Rho-kinase) are Ser/Thr-protein kinases that regulate cellular cytoskeletal organization downstream of the small GTPase Rho. ROCKs are broadly expressed in tissues, and aberrant activation of ROCKs has been reported to be involved in various diseases, including cardiovascular diseases, neurodegenerative diseases, ocular diseases, and tumors [10,11,12]. Many inhibitors of ROCKs have been developed, and Fasudil and Ripasudil are clinically used for cerebral vasospasm after surgery of subarachnoid hemorrhage and glaucoma, respectively [13]. Rho-kinase inactivates myosin light chain (MLC) phosphatase (MLCP) through phosphorylating the myosin phosphatase targeting subunit 1 (MYPT1) at Ser696/Thr697/Ser854/Thr855, which leads to the enhancement of MLC phosphorylation [14,15,16]. Rho-kinase also phosphorylates MLC and, thus, directly and indirectly regulates the phosphorylation level of MLC and the resultant cellular contractility [14,16]. It has been reported that the C-terminus of MYPT1 containing Rho-kinase phosphorylation sites interacts with Rho-kinase [17]. We also showed that MYPT1 and Rho-kinase were associated through KISS analysis, using the catalytic fragment of Rho-kinase [7]. Here, we identified the three regions of MYPT1 (704–715 aa, 836–849 aa, and 684–691 aa), which are referred to as docking motif (DM) 1, 2, and 3, respectively, that are responsible for the interaction with the catalytic fragment of Rho-kinase. The MYPT1 fragment containing DMs and phosphorylation sites inhibited Rho-kinase activity, both in vitro and in vivo, but deleting the DMs abolished the inhibitory effect. Synthetic peptides containing DMs and phosphorylation sites replaced by Ala also inhibited the Rho-kinase activity in vitro. The lack of DMs in MYPT1 only resulted in a decrease in phosphorylation at neighboring phosphorylation sites. These results suggest that the DMs of MYPT1 contribute to the efficiency and specificity of phosphorylation by Rho-kinase.

## 2. Materials and Methods

### 2.1. Materials and Chemicals

Anti-GFP monoclonal mouse antibody (Roche Diagnostics, Basel, Switzerland), anti-MYPT1-pT853 (pT855 in Rat) rabbit polyclonal antibody (EMD Millipore, Billerica, MA, USA), anti-MYPT1-pT696 (pT697 in Rat) rabbit polyclonal antibody (EMD Millipore, Billerica, MA, USA), and Alexa555-Phalloidin (Thermo Fisher Scientific, Waltham, MA, USA) were purchased. Anti-GST polyclonal rabbit antibody and anti-MYPT1-pT854 polyclonal rabbit antibody were produced against recombinant GST and phosphopeptide CREKRRpSTGVSF, respectively. cDNAs encoding bovine Rho-kinase (gene name ROCK2, UniProt Q28021-1), rat MYPT1 (gene name Ppp1r12a, UniProt Q10728-1), human ROCK1 (UniProt Q13464-1), and human MRCKβ (gene name CDC42BPB, UniProt Q9Y5S2-1) were cloned from cDNA libraries, and subcloned into pEF-BOS-GST [18], pEGFP-c1 (Takara Bio Inc., Shiga, Japan), or pGEX-2T (Cytiva, Marlborough, MA, USA) expression vector. GST-Rho-kinase-cat-L (bovine, residues 6–553), GST-ROCK1-cat-L (human, residues 1–536), and GST-MRCKβ-cat-L (human, residues 1–550) were expressed in Sf9 cells using a baculovirus system. Recombinant GST-PKACα (Carna Bioscience, Kobe, Japan) was purchased.

### 2.2. Protein Preparation

GST-recombinant kinases were purified from Sf9 cells using a baculovirus expression system [19]. The cells were lysed in lysis buffer (25 mM Tris-HCl pH 7.5, 2 mM EGTA, 1 mM DTT, 150 mM NaCl, 0.25 M sucrose, 20 mM β-glycerophosphate, 20 mM NaF, 100 μM APMSF, 2 μg/mL leupeptin) by sonication. Recombinant kinases were purified with Glutathione Sepharose beads (Cytiva, Marlborough, MA, USA), eluted with 10 mM glutathione, and dialyzed in buffer (25 mM Tris-HCl pH 7.5, 2 mM EGTA, 1 mM DTT). GST-MYPT1 fragments with or without mutations were purified from E. coli (BL21/pLys). The cells were lysed in lysis buffer (20 mM Tris-HCl pH 7.5, 1 mM EDTA, 1 mM DTT, 0.25 M sucrose, 100 μM APMSF, 2 μg/mL leupeptin) by sonication. Recombinant proteins were purified with Glutathione Sepharose beads, eluted with 10 mM glutathione, and dialyzed in buffer (20 mM Tris-HCl pH 7.5, 1 mM EDTA, 1 mM DTT).

### 2.3. Pull-Down Assay

COS7 cells were seeded on a 6-well dish at a density of 1.8 × 10^5^ cells/well in Dulbecco’s modified Eagle’s medium (DMEM) with 10% fetal bovine serum and cultured overnight. The cells were transfected with plasmids using LIPOFECTAMINE 2000^TM^ reagent (Thermo Fisher Scientific, Waltham, MA, USA), according to the manufacture’s protocol. Glutathione Sepharose beads were incubated with COS7 cell lysates transfected with the indicated plasmids in lysis buffer (20 mM Tris-HCl pH 7.5, 1 mM EDTA, 1 mM DTT, 50 mM NaCl, 0.5% NP-40, 0.1 mM APMSF, 0.5 μg/mL aprotinin, and 2 μg/mL leupeptin) for 1 h at 4 °C, and the beads were subsequently washed three times with lysis buffer. After washing, the beads were resuspended in SDS–PAGE sample buffer, and the bound proteins were subjected to immunoblot analysis using the indicated antibodies. The signals were visualized using HRP-conjugated secondary antibody and ECL prime detecting reagent (Cytiva, Marlborough, MA, USA) by X-lay film or imaging analyzer LAS 4010 (Cytiva, Marlborough, MA, USA).

### 2.4. Analysis of Phosphorylation of GFP-MYPT1 in COS7 Cells

COS7 cells were seeded on a 6-well dish at a density of 1.8 × 10^5^ cells/well in DMEM with 10% fetal bovine serum and cultured overnight. The cells were transfected with plasmids using LIPOFECTAMINE 2000^TM^ reagent, according to the manufacture’s protocol. After 24 h, the cells were washed twice with PBS, lysed in SDS–PAGE sample buffer and, subjected to immunoblot analysis using the indicated antibodies. The signals were visualized using HRP-conjugated secondary antibody and ECL prime detecting reagent (Cytiva, Marlborough, MA, USA) by X-lay film or imaging analyzer LAS 4010 (Cytiva, Marlborough, MA, USA). Signal intensities of phospho-GFP-MYPT1 and GFP-MYPT1 were analyzed using Fiji software [20] and quantified from standard curves. The phospho-GFP-MYPT1 levels were normalized by GFP-MYPT1 levels.

### 2.5. In Vitro Kinase Assay

The kinase reactions were performed in 50 μL of a reaction mixture (25 mM Tris-HCl, pH 7.5, 1 mM EDTA, 1 mM EGTA, 1 mM DTT, 5 mM MgCl_2_, and 50 μM γ-[^32^P]ATP [1–20 GBq/mmol]), GST-Rho-kinase-cat-L or other kinases (1–10 nM), and substrates for 10–20 min at 30 °C. The reaction mixtures were applied to cation-exchange membranes and washed with 75 mM phosphoric acid three times, which was followed by scintillation counting.

### 2.6. Immunocytochemistry

NIH3T3 cells were seeded on polylysine-coated 13-mm coverslips at a density of 4 × 10^4^ cells/well in a 6-well dish and cultured in DMEM with 10% calf serum overnight. The cells were transfected with plasmid using LIPOFECTAMINE LTX reagent (Thermo Fisher Scientific, Waltham, MA, USA), according to the manufacture’s protocol. After 24 h, the cells were incubated in fresh DMEM containing 10% calf serum for 20 min, fixed with 3.7% formaldehyde in PBS for 10 min and then treated with PBS containing 0.2% Triton X-100 for 10 min. The fixed cells were blocked with 1% bovine serum albumin for 30 min and stained with the indicated antibodies. Cover slips were mounted with Fluoromount (Diagnostic Biosystems, Pleasanton, CA, USA) and the fluorescence was examined using a confocal laser scanning microscope LSM 780 and LSM5 PASCAL built around an Axio Observer Z1 or Axiovert 200M under the control of LSM software (Carl Zeiss, Oberkochen, Germany). Images were processed using ImageJ or Photoshop (Adobe, San Jose, CA, USA).

### 2.7. Statistics

Prism (version 6.0, GraphPad Software) was used for statistical analysis. Data were evaluated using one-way or two-way analysis of variance followed by a post hoc Dunnett’s or Sidak’s multiple comparison test.

## 3. Results

### 3.1. Interaction of MYPT1 with Rho-Kinase

MYPT1 was reported to interact with Rho-kinase (354–775 aa) through its C-terminal region that contains phosphorylation sites by Rho-kinase [17]. We also found that the catalytic fragment of Rho-kinase (Rho-kinase-cat-L; 6–553 aa, long form of catalytic fragment) bound to the MYPT protein family [7]. To confirm the interaction of Rho-kinase with MYPT1, GST-MYPT1 fragments were expressed in COS7 cells along with GFP-Rho-kinase-cat-L and were pulled-down using glutathione beads. As reported, Rho-kinase-cat-L coprecipitated with the C-terminal fragment of MYPT1 (592–1032 aa), but not the N-terminal (1–300 aa) or central (296–597 aa) fragment (Figure 1A,B). To further narrow the binding region, the ability of various MYPT1 fragments to bind Rho-kinase-cat-L was examined. Wang et al. demonstrated that the 683–866 aa region of human MYPT1 containing four major phosphorylation sites (corresponding to 684–868 aa of rat MYPT1 in this study) interacted with Rho-kinase [17]. Rho-kinase-cat-L consistently coprecipitated with the 684–868 aa region of MYPT1 (Figure 1B). The fragments of 655–715 aa, 758–868 aa, and 713–868 aa, which contain two respective phosphorylation sites, also interacted with Rho-kinase-cat-L. Unexpectedly, the 655–703 aa fragment failed to bind to Rho-kinase-cat-L, even though it contained two phosphorylation sites (Ser696/Thr697) (Figure 1B), suggesting that the 704–715 aa region contributes to the interaction. Replacing the four phosphorylation sites with Ala in the 684–868 aa fragment reduced its ability to bind to Rho-kinase-cat-L, while single or double Ala mutation caused little effect (Figure 1C). The fragment of 684–758 aa with 696A/697A mutations was not able to bind Rho-kinase-cat-L, whereas the fragment of 758–868 aa with 854A/855A mutations showed a similar affinity as that of the wild type (Figure 1C). These results suggest that the MYPT1 684–868 aa region containing four phosphorylation sites interacts with Rho-kinase through phosphorylation sites and that additional regions, other than phosphorylation sites, are also involved in this interaction.

As the Rho-kinase-cat-L fragment includes both catalytic domain and some coiled-coil domain (Appendix A), the respective domains were separately analyzed for MYPT1 binding. The core catalytic fragment (Rho-kinase-cat, 6–418 aa) and part of the coiled-coli fragment (418–553 aa) equally and weakly interacted with the MYPT1-684–868 aa region, respectively (Appendix A). The catalytic fragment of ROCK1 (ROCK1-cat-L), another isoform of Rho-kinase, also interacted with the C-terminus of MYPT1 (Appendix A). As with ROCKs, MRCKβ belongs to the DMPK subfamily of kinases, and the amino acid sequence of the catalytic domain is 46% identical to that of Rho-kinase. Despite the high sequence similarity, MRCKβ-cat-L failed to interact with MYPT1 (Appendix A). The AGC family of kinases, including DMPK and several other groups of kinases, prefers the phosphorylation site surrounded by basic amino acid residues and shares similar consensus sequences [1,2]. We, thus, compared the interaction of the catalytic fragments of other kinases with MYPT1. PKA was shown to have a very weak interaction with MYPT1, but the other kinases did not have a detectable interaction with MYPT1, as tested (Appendix A). The catalytic domain of ROCKs, thus, specifically interacts with MYPT1. 

### 3.2. Identification of the Docking Motif of MYPT1

We hypothesized the presence of a Rho-kinase-interacting docking motif within the MYPT1-684–868 aa region. Since 704–715 aa of MYPT1 seemed to be important for the binding to Rho-kinase (Figure 1B), a MYPT1 fragment missing this region was tested. Deletion of the 704–715 aa region dramatically reduced the interaction between the MYPT1-684–758 aa fragment and Rho-kinase-cat (Figure 2A). A series of deletion mutants of the MYPT1-758-868 aa fragment revealed that the 836–849 aa region, but not the 814–822 aa or 822-835 aa region, was required for the interaction with Rho-kinase-cat (Figure 2A). Here, we refer to the 704–715 aa and 836–849 aa regions of MYPT1 as DM1 and DM2 for Rho-kinase, respectively. Deleting DM1 or DM2 reduced the binding ability of the MYPT1-684–868 aa fragment to Rho-kinase-cat, and a stronger effect was observed with DM1 deletion than DM2 deletion (Figure 2B). Deleting both DM1 and DM2 completely abolished Rho-kinase-cat binding, even though all the phosphorylation sites were intact (Figure 2B), suggesting that the DM1 and DM2 of MYPT1 interact with and recognize Rho-kinase. DM1 and DM2 are located outside of the consensus but are close to the phosphorylation sites at the C-terminal and N-terminal sides, respectively. Apparent sequence homology was not observed between DM1 and DM2. 

### 3.3. Inhibition of Rho-Kinase by the Inhibitory Fragment/Peptide of MYPT1

We also examined whether these Rho-kinase-binding MYPT1 fragments inhibit the interaction between Rho-kinase and MYPT1 and resultant MYPT1 phosphorylation. Full-length of GFP-MYPT1 was coexpressed with various MYPT1 fragments in COS7 cells, as shown in Figure 1A. The MYPT1-684–868 aa fragment strongly inhibited the phosphorylation of GFP-MYPT1 at Thr855, to a similar extent as that of dominant negative (DN) Rho-kinase (RB/PH(TT); autoinhibitory region with Rho-binding-deficient mutations) (Figure 3A). Even though the expression levels of potential inhibitory MYPT1 fragments were varied, especially in Figure 3A, and the inhibitory effect of each fragment could not be directly compared, the inhibitory effects on GFP-MYPT1 phosphorylation by the MYPT1 fragments largely correlated with their ability to bind Rho-kinase-cat, with several exceptions (Figure 3). The MYPT1-684–758 aa-696A/697A fragment failed to interact with Rho-kinase-cat (Figure 1C) but inhibited MYPT1 phosphorylation (Figure 3B). In contrast, the MYPT1-758–868 aa-854A/855A fragment markedly interacted with Rho-kinase-cat (Figure 1C) but failed to inhibit MYPT1 phosphorylation (Figure 3B). These fragments may indirectly influence MYPT1 phosphorylation. In addition, similar effects of these fragments on endogenous MYPT1 phosphorylation were observed (Figure 3A,B). Under our experimental conditions, the transfection efficiency was about 30~40% judged from GFP fluorescence, which means that 60~70% of endogenous MYPT1 did not coexist along with exogenous proteins and that the apparent effects of exogenous inhibitory proteins on endogenous MYPT1 are thought to be weakened. These results suggest that the Rho-kinase-binding MYPT1 fragments interfere with the phosphorylation of both exogenous GFP-MYPT1 and endogenous MYPT1 by inhibiting endogenous Rho-kinase. 

The inhibitory effects of MYPT1 fragments on actin stress fiber formation were examined in NIH3T3 cells, to determine whether these fragments actually inhibit Rho-kinase activity in cells. Expression of the MYPT1-684-868 aa or -684-868 aa-4A fragment reduced actin stress fibers, and the effect was milder than that of DN-Rho-kinase (Figure 3C). In contrast, expression of the MYPT1-655-715 aa fragment rather enhanced stress fiber formation (Figure 3C), although it decreased the phosphorylation of MYPT1 in COS7 cells (Figure 3A). As Khromov et al. previously reported that human MYPT1-654–714 aa and -654–880 aa fragments in the phosphorylated forms inhibit MLCP activity in vitro and induce the smooth muscle contraction [21], this fragment might exert a stronger effect on MLCP than that of Rho-kinase. These results suggest that the MYPT1-684–868 aa fragment suppresses endogenous Rho-kinase activity through interacting with Rho-kinase. 

To examine whether the MYPT1 fragment directly inhibits Rho-kinase activity, an in vitro kinase assay was performed using the S6 rsk substrate peptide as a substrate. The MYPT1-684–868 aa fragment inhibited the Rho-kinase-cat-L activity in a dose-dependent manner (Figure 4A). MYPT1 fragment itself was also phosphorylated under these conditions. The MYPT1-684–868 aa-4A fragment was barely phosphorylated by Rho-kinase and inhibited the Rho-kinase-cat-L activity (Figure 4A). Similar inhibitory effects on ROCK1-cat-L activity by MYPT1 fragments were observed (Figure 4A). In addition to the S6 rsk substrate peptide, phosphorylation of MLC and Scrib, other already known substrates for Rho-kinase, was also inhibited by the MYPT1-684–868 aa-4A fragment (Appendix A). The inhibitory effects of the MYPT1-684–868 aa-4A fragment on MYPT1 phosphorylation were much weaker (Appendix A), possibly because MYPT1 has a higher affinity for Rho-kinase than MLC and Scrib. These results indicate that the MYPT1-684–868 aa fragment directly and generally inhibits Rho-kinase activity. MYPT1-684–868 aa-4A did not inhibit the PKA activity (Appendix A), but unexpectedly, it moderately inhibited the activities of MRCKβ (Appendix A). In an in vitro kinase assay, the GST-MYPT1-684–868 aa-4A fragment lacking DM1 or DM2 exerted a weaker inhibitory effect on the Rho-kinase-cat-L activity, and the absence of both DM1 and DM2 resulted in no inhibitory effect (Figure 4B). These results indicate that the DM1 and DM2 of MYPT1 are necessary for the inhibition of Rho-kinase.

Since the kinase–substrate interfaces serve as the target of peptide inhibitors such as pseudosubstrate (PS) and docking site/motif, synthesized peptides derived from DM1, DM2, and sequences surrounding phosphorylation sites with alanine substitution (Figure 4C and Table 1) were analyzed as potential Rho-kinase inhibitors. Using the S6 rsk substrate peptide as a substrate, the Rho-kinase activity was strongly inhibited by the PS2 + DM2 peptide and moderately inhibited by the DM2 peptide but not by any other peptide (Figure 4D). Since the PS2 + DM2 peptide showed a stronger inhibitory effect than that of the PS2 or DM2 peptide, the PS2 + DM2 peptide is thought to interfere with the kinase–substrate interaction through binding to Rho-kinase with a much higher affinity than that of PS2 or DM2. Unexpectedly, PS1 + DM1 peptide did not show an inhibitory effect on Rho-kinase activity, and a longer sequence or ternary structure may be necessary for interacting with Rho-kinase. The PS2 + DM2 peptide also inhibited the phosphorylation of MYPT1, MLC, and Scrib by Rho-kinase (Appendix A). 

We further searched for the MYPT1 sequence necessary for the interaction with Rho-kinase, because the PS1 + DM1 peptide failed to suppress Rho-kinase activity, even though the 684–758 aa fragment reduced MYPT1 phosphorylation. We found that a basic amino acid cluster exists (683–691 aa), similar to DM2, at the N-terminus of Ser696/Thr697 (Figure 4C) and that deleting 683–691 aa reduced the binding ability of the MYPT1 fragment to Rho-kinase (Appendix A). This 683–691 aa region was thought to be the third docking motif and is hereafter referred to as DM3 (Figure 4C). The synthetic peptide PS1 + DM3 (683–703 aa), but not the DM3 peptide, inhibited Rho-kinase activity to a similar extent as PS2 + DM2 in vitro (Figure 4E). 

### 3.4. Effects of Docking Motifs on the Phosphorylation of MYPT1

To evaluate the effects of docking motifs on MYPT1 phosphorylation, GST-MYPT1-684–758 aa and -758–868 aa fragments with or without DMs were purified from *E. coli* and subjected to in vitro phosphorylation analyses. Deletion of DM1 or DM3 from the 684–758 aa fragment did not alter the phosphorylation of each fragment by Rho-kinase, whereas deleting both DM1 and DM3 almost completely abolished phosphorylation (Figure 5A). Deleting DM2 from the 758–868 aa fragment completely reduced phosphorylation (Figure 5A). In COS7 cells, the full-length of GFP-MYPT1 lacking DM1, DM2, and/or DM3 was expressed, and the phosphorylation levels at Thr855, Ser854, and Thr697 were examined. Deleting both DM1 and DM3 decreased phosphorylation at Thr697 but not Ser854 or Thr855, whereas the deletion of DM2 decreased phosphorylation at Thr855 and Ser854 but not Thr697 (Figure 5B). These results suggest that each DM affects phosphorylation only at neighboring sites and does not influence the distant sites. 

### 3.5. Kinase-Substrate Interaction of Rho-Kinase

To assess the interface of Rho-kinase with MYPT1, Rho-kinase–MRCKβ chimera mutants of the catalytic domain were investigated because MRCKβ is closely related to Rho-kinase but fails to interact with MYPT1 (Appendix A). Based on previous studies [7,17], the C-terminus of the Rho-kinase catalytic domain was thought to be mapped as a MYPT1-binding region. The chimera mutants largely lost the ability to bind to MYPT1 (Appendix A). Replacing the C-terminal 19 aa (400–418 aa) of Rho-kinase, which corresponds to the hydrophobic motif necessary for AGC family kinase activity [22,23], resulted in the complete loss of interaction with MYPT1 (Appendix A), even though the essential FXXF[T/S][Y/F] sequence within the hydrophobic motif was conserved (Appendix A). Nonetheless, because MRCKβ with the C-terminus of Rho-kinase (400–418 aa) did not restore the binding ability, this C-terminal region is thought to be necessary but not sufficient for the interaction with MYPT1. Analyses of the mutants in which one or two amino acid residues of the C-terminus of the Rho-kinase catalytic domain were replaced by those of MRCKβ revealed that alteration of amino acid residue(s) within the 396–407 aa region of Rho-kinase decreased the association with MYPT1 (Appendix A), suggesting this region is important as a potential docking site. The kinase activities of these mutants were also reduced to varying degrees (Appendix A).

We also surveyed similar sequences to MYPT1 DMs in the already-known Rho-kinase substrates. Basic amino acid clusters, such as DM2 and DM3, are found near the phosphorylation sites of Scrib, MLC, ARHGAP35, MARCKS, and Adducin1. Among them, we confirmed the sequence rich in basic amino acids of Scrib (1451-PLGGGAPVRTAKAERRHQERLRVQSPE-1477) close to the phosphorylation site Ser1508 was involved in the Rho-kinase binding (Appendix A), suggesting that the poly basic sequence widely serves as a docking motif for Rho-kinase.

## 4. Discussion

### 4.1. Rho-Kinase-MYPT1 Interaction

The MYPT1 of MLCP is one of the physiologically important substrates for Rho-kinase, which broadly regulates cellular contractility [16]. In this study, we found that phosphorylation sites (Ser696/Thr697/Ser854/Thr855) and three neighboring regions (docking motifs; DM1 704–715 aa, DM2 836-849 aa, and DM3 683–691 aa) of MYPT1 contribute to the interaction with, and phosphorylation by, the catalytic region of Rho-kinase. The MYPT1 fragment containing all these components (684–868 aa) showed high affinity and preferentially associated with both Rho-kinase and ROCK1, in comparison with other kinases (Appendix A). Replacing phosphorylation sites to Ala or deleting DMs from the MYPT1 fragment reduced the binding ability, and a lack of all components resulted in a loss of binding to Rho-kinase. Phosphorylation sites directly interact with the active center, and DMs are thought to associate with the potential docking site(s) of Rho-kinase within the catalytic domain (Figure 6B). MYPT1 fragments, thus, competed with substrates and exerted inhibitory effects on the kinase activity of Rho-kinase, of which the efficiencies were almost correlated with its binding abilities. In COS7 cells, the lack of DMs in MYPT1 resulted in a reduction in phosphorylation of only neighboring sites but not distant sites, suggesting that each DM of MYPT1 increases the local affinity of the phosphorylation site for the active center of Rho-kinase. 

The C-terminus of the Rho-kinase catalytic domain was thought to be the interface for MYPT1, even though it was not sufficient. Replacing the amino acid of Rho-kinase within the 396–407 aa region by MRCKβ decreased the binding to MYPT1 (Appendix A) and kinase activity (Appendix A). This region is included in the C-terminal extension of the Rho-kinase catalytic domain. Rho-kinase, as well as other DMPK family members, has been reported to form a head-to-head homodimer through the N-terminal and C-terminal extensions of the catalytic domain, which is required to keep its active form [22]. We also found that the Rho-kinase Asp58Ser mutant within the N-terminal extension also decreased both the binding to MYPT1 and the kinase activity (unpublished observation); therefore, the dimerization interface might be the docking site to MYPT1 (Appendix A). Interaction between the DMs of MYPT1 and the docking site(s) of Rho-kinase is anticipated to enhance the efficiency and specificity of vicinal phosphorylation events. 

### 4.2. Docking Motifs of MYPT1

The docking interaction of the MAPKs with their substrates and regulators has been well studied. At least two types of docking motifs for MAPKs have been reported: D-motif (also known as KIM-motif) and FXFP-motif, which interact with the different sites of the catalytic domain of MAPKs [1,5,24]. The D-motif is an 8–15 aa sequence satisfying loose consensus (R/K)_2–3_-X_2–6_-Φ-X-Φ (Φ is any hydrophobic residue, and both the N to C and C to N directions are acceptable), with varying distances from the phosphorylation site. The basic and hydrophobic amino acids of the D-motif bind to the acidic patch and hydrophobic groove of MAPKs, respectively. Here, we identified the DM1 (DLQEAEKTIGRS), DM2 (KSQPKSIRERRRPR), and DM3 (RKARSRQAR) sequences of MYPT1, but these sequences are not highly conserved. DM2 and DM3 are rich in basic amino acids, and both are located at the N-terminus of the respective phosphorylation sites (Figure 4C). Replacing basic amino acids within DM2 or DM3 with Ala, instead of deleting DM2 or DM3, also interfered with the Rho-kinase association (Appendix A), suggesting polar interactions exist with Rho-kinase through these motifs. DM1 is different from DM2 and DM3 and rather rich in acidic amino acids (Figure 4C), implying that this motif interacts with different sites of Rho-kinase. Deleting DM2 was sufficient to result in the loss of phosphorylation at Ser854/Thr855, whereas the deletion of both DM1 and DM3 was necessary to abolish phosphorylation at Ser696/Thr697 (Figure 5). The affinity and/or mode of interaction of each docking motif may be divergent. 

The solution structure of human MYPT1-658–714 aa based on the NMR spectroscopy and structural calculation has been reported, in which the MYPT1-658–714 aa consists of three helixes (A, B, and C) (Appendix A) [25]. DM1 almost corresponds to helix C, and DM3 is located in the center of helix B. The deletion of DM3 may disrupt the helical structure of this region. In the secondary structure prediction of MYPT1-684–868 aa region, DM1 and DM3 are speculated to be alpha helixes, as with the solution structure, whereas DM2 is speculated to be a turn-helix structure (Appendix A). As we cannot exclude the possibility that deletion/mutation in our study denatured and/or destabilized the proper MYPT1 protein structure and consequently reduced the affinity for Rho-kinase, further structural analyses will be necessary to understand the MYPT1 Rho-kinase interface(s) in future.

The DMs identified in this study are completely conserved between rat and human MYPT1 (Gene symbol: PPP1R12A). We were intrigued by mutations in these regions and the potential involvement with diseases in human MYPT1. In dbSNP, we found nine, six, and one human missense SNPs in DM1, DM2, and DM3, respectively. Among these missense SNPs, rs1872606231 (E708[709]G), rs1565744239 (I711[712]V), rs1395690034 (G712[713]R), rs1872428296 (S839[841]T), and rs1872426757 (R847[849]G) may alter the affinity of MYPT1 to Rho-kinase (numbers in [ ] indicates amino acid number in rat) (Appendix A). Though there is no description of the pathogenicity or clinical report for these SNPs, possibly due to the very low frequency, we will pay continuing attention to mutations in DMs.

In human proteins, sequences of three DMs completely match the corresponding sequences of human MYPT1, as mentioned above, and closely match those of human MYPT2 (Gene symbol PPP1R12B; DM1 75%, DM2 64%, DM3 100% identical) and MBS85 (Gene symbol PPP1R12C; DM1 50%, DM3 78% identical). Other than the MYPT family members, we cannot find any protein with high similarity to MYPT1 DMs, though some proteins have sequences with partial similarities. Due to the low sequence similarity among the DMs of MYPT1, prediction of the phosphorylation site of substrates by Rho-kinase, especially from the huge shotgun phosphoproteomic data, still seems arduous. We searched for the similar sequences in the already-known Rho-kinase substrates. Basic amino acid clusters, such as DM2 and DM3, are found near the phosphorylation sites of Scrib, MLC, ARHGAP35, MARCKS, and Adducin1. The sequence rich in basic amino acids of Scrib (1451-PLGGGAPVRTAKAERRHQERLRVQSPE-1477) close to the phosphorylation site Ser1508 was confirmed to be involved in the Rho-kinase binding (Appendix A). In addition, among the 365 candidate phosphorylation sites by Rho-kinase detected in KISS analysis [7], 69 sites satisfied ‘poly basic within −15~+15 aa position and phosphorylation consensus’. These results suggest that the poly basic sequence is one of the common docking motifs for Rho-kinase. More instances of docking motifs for Rho-kinase, as with MAPKs, would likely improve the precise prediction of phosphorylation sites by Rho-kinase, which would lead to an understanding of the whole phosphorylation network. 

### 4.3. Inhibitors of Rho-Kinase Derived from Docking Motifs

Pseudosubstrate peptides have been utilized as inhibitors for certain kinases, such as PKA, PKC, and CaMKII, which contain consensus sequences with substitution of Ala for phosphorylation sites [1,2,6]. Peptide inhibitors derived from docking sites and docking motifs have also been developed for the MAPK family of kinases [6]. Recently, Qvit et al. demonstrated the substrate-specific phosphorylation inhibition by PKCδ using peptides of predicted docking motifs of MARCKS, Drp1, and IRS1 [26]. We anticipated that DM peptides of MYPT1 act as MYPT1-specific inhibitors against Rho-kinase. Unfortunately, DM peptides, as well as PS peptides, exerted little or weak inhibitory effect on Rho-kinase activity (Figure 4D,E). In contrast, the PS2 + DM2 and PS1 + DM3 peptides efficiently reduced the Rho-kinase activity (IC50 < 1 μM), possibly because the affinity of each peptide for Rho-kinase was synergistically increased through the dual contact sites. These PS + DM peptides are thought to be general Rho-kinase inhibitors that mask the active center of kinase. Actually, the PS2 + DM2 peptide suppressed the phosphorylation of not only MYPT1 but also the S6 rsk substrate peptide, MLC, and Scrib (Appendix A). The PS1 + DM1 peptide showed a mild inhibitory effect at a higher concentration (100 μM) (data not shown), suggesting a weaker affinity than that of the PS2 + DM2 and PS1 + DM3 peptides. In addition, these sequences with high affinity and specificity for Rho-kinase would also be applicable to the improvement of cellular monitoring tools, such as FRET biosensors for Rho-kinase [27]. 

In summary, we identified novel interfaces (DMs) of MYPT1 with Rho-kinase and found that these DMs contribute to efficient and specific phosphorylation at the proximal sites. Synthetic peptides containing both DM and PS sequences inhibited the activity of Rho-kinase and can serve as potential Rho-kinase-specific inhibitors. Information on docking motifs with consensus sequences will enable us to predict the phosphorylation site of Rho-kinase.

## Figures and Tables

**Figure 1 biomolecules-12-00159-f001:**
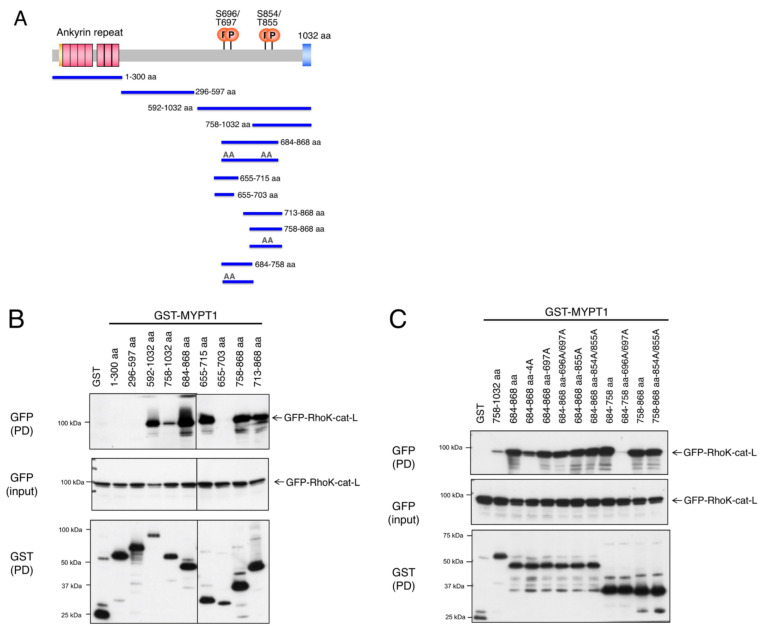
Interaction of MYPT1 with Rho-kinase. (**A**) Schematic representation of the domain structure and deletion constructs of MYPT1. (**B**,**C**) Mapping of the Rho-kinase-interacting region of myosin phosphatase targeting subunit 1 (MYPT1). COS-7 cells were cotransfected with the GST-MYPT1 fragment and GFP-Rho-kinase-cat-L (6–553 aa) and pulled-down with glutathione beads. The bound proteins were subjected to immunoblot analysis using an anti-GST or anti-GFP antibody. The C-terminus of MYPT1 bound to Rho-kinase-cat-L, in which phosphorylation sites of MYPT1 contributed to the binding. These results are representative of at least three independent experiments.

**Figure 2 biomolecules-12-00159-f002:**
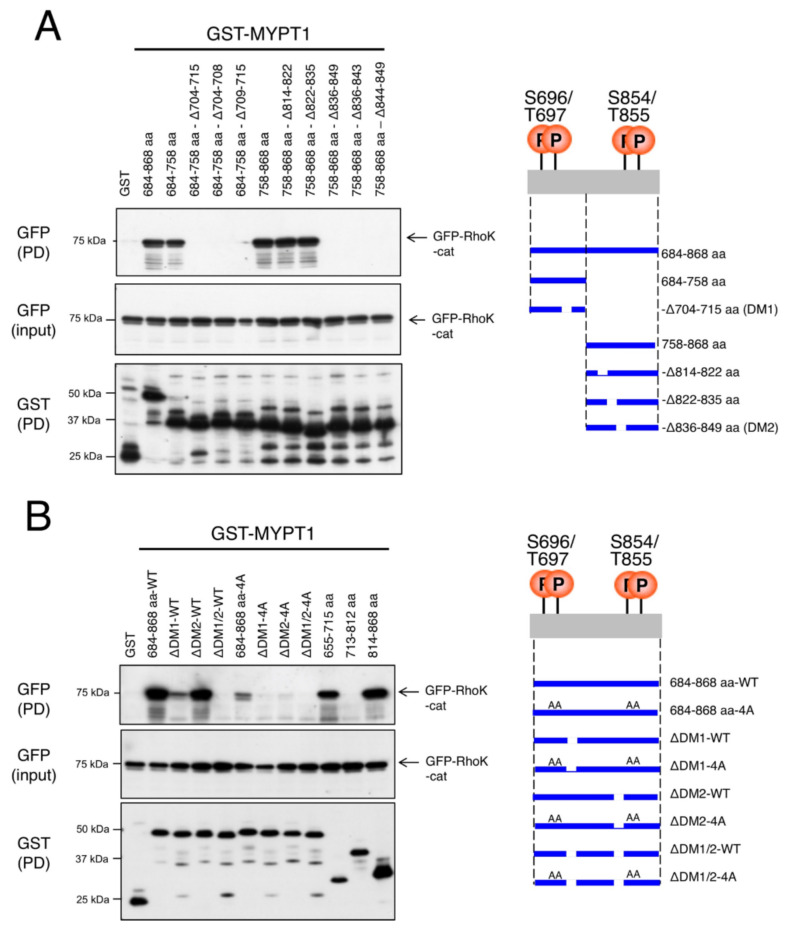
Identification of the docking motifs of MYPT1. (**A**,**B**) Identification of the regions/sites responsible for the binding to Rho-kinase. COS-7 cells were cotransfected with the GST-MYPT1 fragment with deletion or amino acid substitution and GFP-Rho-kinase-cat (6–418 aa) and pulled-down with glutathione beads. The bound proteins were subjected to immunoblot analysis using an anti-GST or anti-GFP antibody. The deletions of docking motif (DM)1 (704–715 aa) and DM2 (836–849 aa) and/or substitutions of phosphorylation sites diminished the association with Rho-kinase. These results are representative of at least three independent experiments.

**Figure 3 biomolecules-12-00159-f003:**
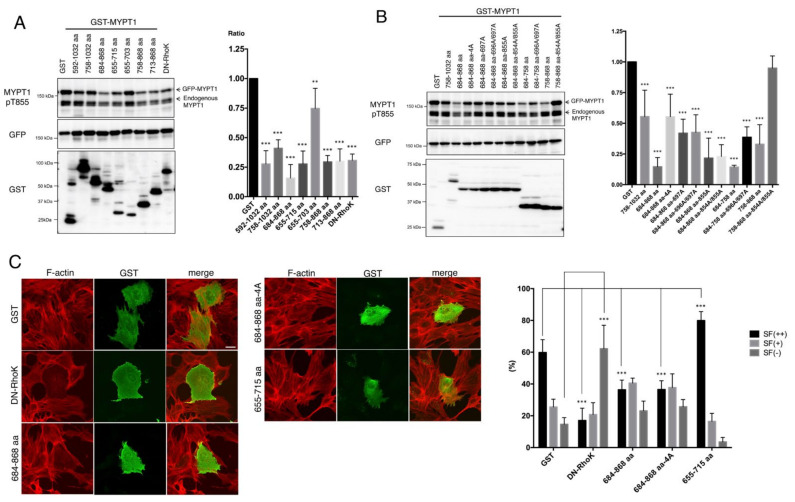
Inhibition of MYPT1 phosphorylation by the Rho-kinase-binding fragment of MYPT1. (**A**,**B**) Effect of the overexpression of Rho-kinase-binding MYPT1 fragments on GFP-MYPT1 phosphorylation in COS7 cells. Full-length GFP-MYPT1 and the GST-MYPT1 Rho-kinase binding fragment were transiently expressed in COS7 cells. The cell lysates were analyzed by immunoblot analysis using anti-MYPT1-pT855 (top panel), anti-GFP (middle), or anti-GST (bottom) antibodies. The phospho-GFP-MYPT1 levels were quantified and normalized by GFP-MYPT1 levels. Data represent means ± SD. **, *p* < 0.01, ***, *p* < 0.001 as compared with the control. (**C**) Effect of the overexpression of Rho-kinase-binding MYPT1 fragments on stress fiber formation in NIH3T3 cells. GST-MYPT1 fragments were expressed in NIH3T3 cells. The cells were fixed in formaldehyde, and immunostained with phalloidin and anti-GST antibody. Colors indicate GST (green) and F-actin (red). Scale bar, 10 μm. The percentages of cells with stress fibers throughout the cell (SF(++)), cells with stress fibers partly in the cell (SF(+)), and cells without stress fibers (SF(–)) were analyzed. Data are means ± SD. N = 4. ***, *p* < 0.001 as compared with control. DN-RhoK: dominant negative Rho-kinase.

**Figure 4 biomolecules-12-00159-f004:**
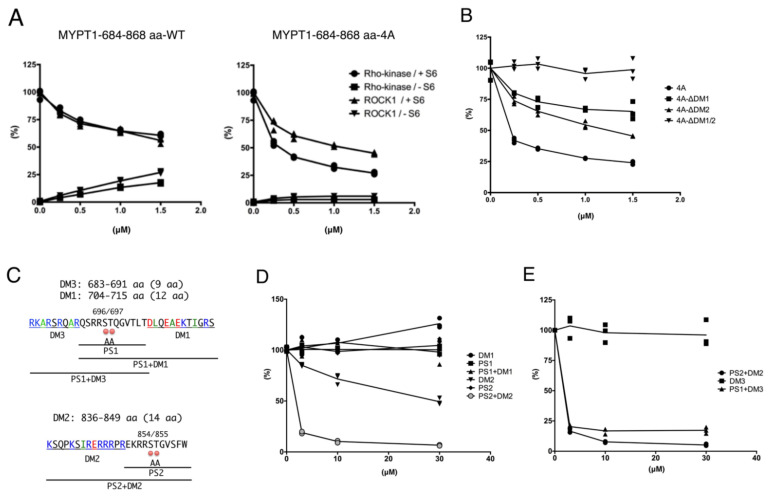
Direct inhibition of Rho-kinase activity in vitro. (**A**) In vitro phosphorylation of S6 rsk substrate peptide by Rho-kinase. GST-Rho-kinase-cat-L or GST-ROCK1-cat-L activity was examined using the S6 rsk substrate peptide as a substrate in the presence of γ-[^32^P]ATP. GST-MYPT1-684–868 aa-WT or -4A was added at the indicated concentrations. The reaction mixtures were applied onto cation-exchange membranes, followed by scintillation counting. N = 3. (**B**) Effects of the deletion of DMs on the inhibition of Rho-kinase activity in vitro. GST-Rho-kinase-cat-L activity was examined using the S6 rsk substrate peptide as a substrate in the presence of γ-[^32^P]ATP. GST-MYPT1-684–868-4A with or without DMs was added at the indicated concentrations. The reaction mixtures were applied onto cation-exchange membranes, followed by scintillation counting. N = 3. The deletion of DM1 or DM2 moderately weakened the inhibitory effect and the deletion of both DM1 and DM2 completely abrogated the inhibitory effects of the MYPT1 fragment on the Rho-kinase activity. (**C**) Sequences of the DMs and phosphorylation sites of MYPT1. The sequences of MYPT1 683–715 aa and 836-860 aa and synthetic peptides are shown. Basic (blue), acidic (red), and aliphatic (green) amino acids within DMs are indicated. Phosphorylation sites are replaced by Ala in pseudosubstrate (PS) peptides. (**D**) Inhibition of Rho-kinase activity by PS2 + DM2 synthetic peptide. GST-Rho-kinase-cat-L activity was examined using the S6 rsk substrate peptide as a substrate in the presence of γ-[^32^P]ATP. Synthetic peptides were added at the indicated concentrations. The reaction mixtures were applied onto cation-exchange membranes, followed by scintillation counting. N = 3. (**E**) Inhibition of Rho-kinase activity by PS1 + DM3 synthetic peptide. GST-Rho-kinase-cat-L activity was examined using the S6 rsk substrate peptide as a substrate in the presence of γ-[^32^P]ATP. Synthetic peptides were added at the indicated concentrations. The reaction mixtures were applied onto cation-exchange membranes, followed by scintillation counting. N = 3.

**Figure 5 biomolecules-12-00159-f005:**
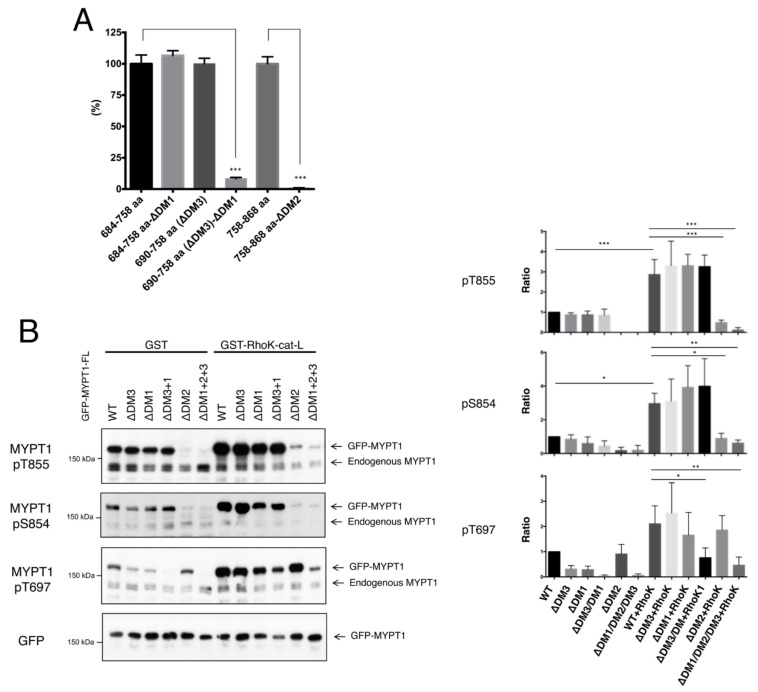
Effects of the DMs on MYPT1 phosphorylation. (**A**) Suppression of MYPT1 fragment phosphorylation by deletion of DMs in vitro. One μM of GST-MYPT1-fragments was phosphorylated by 1 nM of Rho-kinase-cat-L in the presence of γ-[^32^P]ATP. Phosphorylation levels of GST-MYPT1-684–758 aa-ΔDM1, -690–758 aa (deletion of DM3 from 684–758 aa fragment), and -690–758 aa-ΔDM1 (deletion of both DM1 and DM3 from 684–758 aa fragment) were compared with that of GST-MYPT1-684–758 aa, and the phosphorylation level of GST-MYPT1-758-868 aa-ΔDM2 was compared with that of GST-MYPT1-758–868 aa. Data represent means ± SD. N = 3. ***, *p* < 0.001. (**B**) Suppression of MYPT1 phosphorylation by deletion of DMs in COS7 cells. GFP-MYPT1-WT or mutants were transiently expressed with or without GST-Rho-kinase-cat-L in COS7 cells. The cell lysates were analyzed by immunoblot analysis using anti-MYPT1-pT855, anti-MYPT1-pS854, anti-MYPT1-pT697, or anti-GFP antibodies. The phospho-GFP-MYPT1 levels were quantified and normalized by GFP-MYPT1 levels. Data represent means ± SD. *, *p* < 0.05, **, *p* < 0.01, ***, *p* < 0.001 as compared with the control.

**Figure 6 biomolecules-12-00159-f006:**
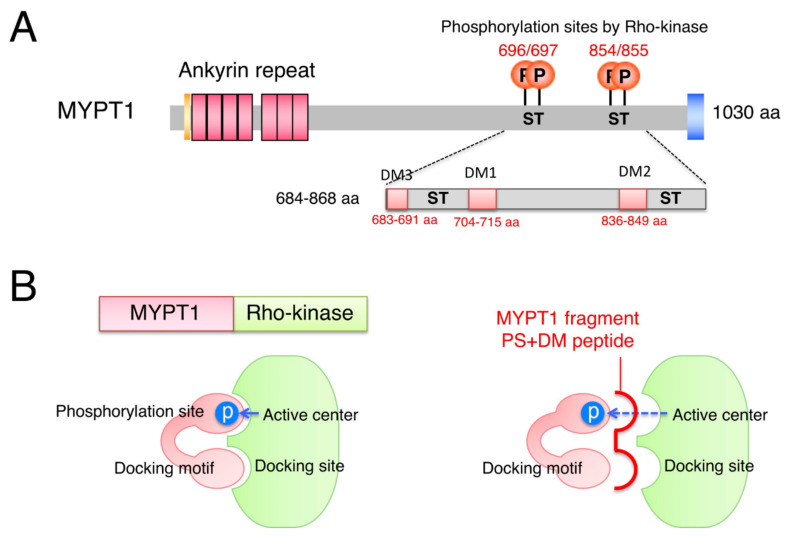
(**A**) Schematic representation of phosphorylation sites and DMs of MYPT1. (**B**) Model for kinase–substrate recognition between Rho-kinase and MYPT1.

**Table 1 biomolecules-12-00159-t001:** Sequences of synthetic peptides.

Peptide	Amino Acid	Sequence
DM1	704–715 aa	DLQEAEKTIGRS
PS1	692–703 aa	QSRRAAQGVTLT
PS1 + DM1	692–715 aa	QSRRAAQGVTLTDLQEAEKTIGRS
DM2	836–849 aa	KSQPKSIRERRRPR
PS2	850–860 aa	EKRRAAGVSFW
PS2 + DM2	836–860 aa	KSQPKSIRERRRPREKRRAAGVSFW
DM3	683–691 aa	RKARSRQAR
PS1 + DM3	683–703 aa	RKARSRQARQSRRAAQGVTLT

## Data Availability

Not applicable.

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
