# Peer review of "Identification of the Kinase-Substrate Recognition Interface between MYPT1 and Rho-Kinase"

_biomolecules, 2022, doi:10.3390/biom12020159_

Round 1

Reviewer 1 Report

The revised Manuscript concerns mapping sites of interaction between MYPT1 protein and Rho-kinase, different than those already known located within  phosphorylation sites and catalytic domain, respectively. Authors prepared  variety of mutated variants of Rho-kinase and its substrate, MYPT1, to find new sites of their interaction and based on  that proposed peptides with inhibitory potential toward Rho-kinase.

The main concern is the quality of signatures/descriptions presented in
the figures. The text is far too small, blurred and illegible, what unable
verification of information stated in the main body of the Manuscript and thus disqualify the Manuscript in current form from being published.

Although the Manuscript regards an interesting issue, especially in the
context of potential application of the research, the scientific soundness is
unsatisfactory and interpretation of some results (those with recognizable  figure signatures) seems inapropriate.

Apart from the major problem with figures clarity there are some other points that should be improved.
The choice of sequence and preparation of truncated mutants of MYPT1 should be justified also by secondary structure inspection (shown in main or
supplementary section) as disruption of some important structural parts of the protein can destabilize whole protein or peptide and also can change its features including pattern of interaction with other molecules, what is important while planning such experiments.

Additionally, in my opinion, some experiments are not conducted totally
properly and lack precise description, eg. Fig.3.  There is an interesting assumption and design but with rather unjustified conclusions. With such different expression level of GST-MYPT1 fragments it is hardly possible to make any quantitative comparisons (additionally, the image of GST(PD) do not macht exactly to the rest of presented WB images). There are also plots in this figure but with no headings and no information about way of normalization applied. The calculation should be clearly explained in the main text or in the figure legend. Moreover the behaviour of endogenous MYPT1 in this conditions should be clearly explicated.
Materials and methods section lacks also some other informations, eg. about preparation and expression of MYPT1 fragments or dephosphomimetic mutants.

My next suggestion regards presentation of the 'model of interaction' shown in Fig. 9.  I encourage to use the appropriate PDB file to generate
figure showing the structural details of Rho-kinase and MYPT1 interaction
along with the cartoon already presented.

I have also comment for Introduction: The characteristics of ROCK kinases
might be extended, especially the information of its isoforms should be
included and  it should be clearly stated, which one is the subject of the
Manuscript. In general, there are places in the text, where the identity of the
kinase is not clear and the Reader can have a problem with understanding whether authors describe the behavior of ROCK2, ROCK1 or Rho-kinase in general (example.: line 61, where Rho-kinase is named ROCK and line68, where the same name is recognized as ROCK2; line 168, 176 etc.). It should be precisely pointed each time, which kinase is analyzed and commented.
Additionally, I think that the Manuscript would catch more readership if there will be a small caption about Rho-kinase-related diseases, what could
strengthen the meaning of potential Rho-kinase inhibitors proposed in this
work. Moreover the importance of the study should be underlined and  
hypothetical usefulness of inhibitors should be discussed.      

There are also some minor issues and parts of the text that have been written chaotically:

Lines: 117: statistics instead of statics
Lines: 123-126 and 186-188 I assume that some single words are missing what indispose understanding of the sentences.
Lines: 158, 163, 167 the "1” is missing in the name of the Rho-kinase
substrate: MYPT1 instead of MYPT
Fig.4C: The same contrast should be applied to all microscopic images

Additionally all abbreviations used, including those in figures , should be
explained.

Reviewer 2 Report

In the present studies, the authors used a traditional molecular biological technique (truncation mutation) to determine the region of binding between Rho-kinase and MYPT1.They have identified a docking motif in MYPT1. The techniques utilized is not that innovative, however, the knowledge obtained from the study may be useful to understand the recognition of kinase-substrate. The manuscript is well written and meet the requirements to be published in the journal.

Minor comment:

1)  The resolutions of most figures are too low (fig 1A... figure 4, figure 5, 6). Better figures should be used.

Author Response

We greatly appreciate the reviewer’s favorable comments and suggestion. According to the reviewer’s suggestion, we thoroughly replaced the figures in the revised manuscript.

Reviewer 3 Report

In this manuscript, the authors carefully studied the kinase-substrate interaction between MYPT1 and Rho-kinase, and suggested that short docking motifs close to phosphorylation sites are required to mediate the phosphorylation of MYPT1 by ROCK. In general, the study was well designed, and indeed the authors did a lot of experiments to support their major findings. A minor but essential revision is needed prior to acceptance. My major concerns are below:

  1. In this study, the pull-down assay was performed only in COS7 cells. I am not sure whether the result is an artifact or true hit. Additional cell lines should be included to ensure the reliability of their study.
  2. If possible, I suggest that authors should search dbSNP or COSMIC mutation databases to find whether mutations changing docking motifs naturally exist, and try to form a link between MYPT1-Rho-kinase interaction and human disease. Additional experiment is not necessary, but such a simple analysis will highlight the importance of this study.
  3. The sequence pattern of the docking motif should be clearly defined and stated in abstract. The authors must analyze more substrates of Rho-kinase to demonstrate whether such a docking motif widely appears in Rho-kinase substrates. 
  4. The authors can obtain Rho-kinase interacting proteins from BioGRID or similar databases, and analyze whether some interacting proteins can be Rho-kinase substrates, and whether these potential substrates contain the docking motif.

Reviewer 4 Report

First of all, this manuscript should be reorganized and condensed to make it more logical, fluent, straightforward and compact. There are in total 9 figures in this manuscript. I suggest the authors to reduce them to 5 or maximum 6 figures, by either merging together some of them or moving some to supplementary data. The authors used the first two figures to show identifying the fragments important for the interaction, then jumped to some functional verification, and then went to Figure 5 to identify the binding regions again. I suggest to show the identification of the interacting regions/fragments directly in the first 2 or 3 figures, followed with the functional verifications. Revise the results accordingly.

Second, the terms or names mess up. The authors should make them consistent. And especially when they (including some abbreviations) first come, the author should explain what they are. I can give some examples. In the manuscript, the authors sometimes wrote Rho-kinase/ROCK2, sometimes Rho-kinase, and sometimes ROCK. Another one, Rho-kinase-cat, judging from the name I thought it’s just pure catalytic domain, but according to the authors, it is a fragment contains both the catalytic domain and part of the coiled-coil domain. As a reader, I get confused with these names. The authors must consider this issue and revise thoroughly to make them consistent and easy to follow.

Third, in Figure 3, the authors tested the inhibition of MYPT1 phosphorylation by Rho-kinase-binding fragments by co-expressing these GST-tagged fragments with GFP-tagged MYPT1 in COS7 cells. Based on the WB results, the pT855 signal intensity of endogenous MYPT1 is comparable to that of the GFP-MYPT1. The authors should use endogenous MYPT1, instead of using exogenous one to reduce potential artifacts. And the authors should also notice the inconsistent signal intensity change of pT855 (especially in Figure 7B) between the endogenous and the exogenous MYPT1s, and explain this.

Round 2

Reviewer 1 Report

In the revised Manuscript Authors introduced some of suggested corrections but there are still important issues that in my opinion need improvement and raised concerns. Especially the experiment described in the Figure 3A and 3B and data obtained from experiments with MYPT1 mutants with destabilized secondary structure needs attention.

The experiment shown in Fig. 3:

The idea that stands behind the experiment (Fig.3A and 3B) and its interpretation is really complicated: the exogenously overexpressed GFP-MYPT1 and fragments of MYPT1 with different level of expression and transfection efficiency in the background of native MYPT1 in the same cells (if co-expressed; but can be expressed individually in different cells) and in un-transfected cells. In this setup getting some reliable information is really difficult and thus results are not very convincing. In more details:

  • An external loading control could be useful. Authors state that the phosphorylation level of GFP-MYPT1 was normalized to the total GFP-MYPT1 level and the question here is if the same membrane was re-probed sequentially with two antibodies (p-MYPT1 T855 and GFP) or two membranes with the same material were developed individually. In file with original figures there are 10 slots on membrane with GFP signal and 11 with p-MYPT1 (first repetition), what raised question about re-probing the same membrane with two antibodies. It seems that line with DN-RhoK is missing in GFP-developed blot (?). Please, explain the difference. Which repetition from original figures file was shown on Fig.3A in the Manuscript?

Additionally, I would recommend here to repeat the analysis with some ‘standard’ loading control suitable and external for the proposed experimental setup or if already presented membranes are preserved and available to re-probe them with antibodies recognizing  ‘neutral’ loading control;

  • To have better view on phosphorylation of endogenous MYPT1 I suggest to additionally use antibodies recognizing the total native MYPT1 and then compare the signals. It is still not ideal because of the influence of native MYPT1 from un-transfected cells but can give additional point of reference.
  • Why membrane with GST antibodies do not align with the rest of membranes?
  • The quantitative WB method should be precisely described.

MYPT1 mutants with deletions:

As Authors have shown some of MYPT1 deletions disrupt its secondary structure and thus the question arise if the lack of phosphorylation in some of these mutants in not just the lack of interaction between the distorted mutant and Rho-kinase.

      Additionally, I still encourage to use PDB files for both analyzed proteins but not to show the interaction of MYPT1 and Rho-kinase but to point on existing structures suggested regions of interactions as in proposed cartoon model. In my opinion using native structures could much better reflect the interaction potential.

Minor:

  • Line 25: please be precise and write exactly which DM of MYPT1 could serve as a potent inhibitor: all of DMs or just one or more and if more, which ones?
  • Unification of abbreviations should be applied: DMs or DM(s);
  • Genetic preparation of phosphomimetic variants could be described;
  • It would be nice to explain ‘DN’ abbreviation from figure 3 in figure legend or in the main text;
  • Some plots still miss their headings;
  • I believe it would be easier for Readers to follow the story about DMs with the information about DM3 placed in the text before the figure and Table with it;
  • I encourage to show all of so called ‘unpublished data’ (line 1059)
  • Lines: 1085-1099 are chaotically written and hard to understand. I suggest to re-write this fragment and underline its sense justifying its presence.

Author Response

Response to Reviewer 1 Comments

Revisions have been made as follows according to the reviewer’s comments. We greatly appreciate the reviewer’s helpful comments and suggestions on the manuscript.

In the revised Manuscript Authors introduced some of suggested corrections but there are still important issues that in my opinion need improvement and raised concerns. Especially the experiment described in the Figure 3A and 3B and data obtained from experiments with MYPT1 mutants with destabilized secondary structure needs attention.

Point 1: The experiment shown in Fig. 3:

The idea that stands behind the experiment (Fig.3A and 3B) and its interpretation is really complicated: the exogenously overexpressed GFP-MYPT1 and fragments of MYPT1 with different level of expression and transfection efficiency in the background of native MYPT1 in the same cells (if co-expressed; but can be expressed individually in different cells) and in un-transfected cells. In this setup getting some reliable information is really difficult and thus results are not very convincing. In more details:

Point 1-1: An external loading control could be useful. Authors state that the phosphorylation level of GFP-MYPT1 was normalized to the total GFP-MYPT1 level and the question here is if the same membrane was re-probed sequentially with two antibodies (p-MYPT1 T855 and GFP) or two membranes with the same material were developed individually. In file with original figures there are 10 slots on membrane with GFP signal and 11 with p-MYPT1 (first repetition), what raised question about re-probing the same membrane with two antibodies. It seems that line with DN-RhoK is missing in GFP-developed blot (?). Please, explain the difference. Which repetition from original figures file was shown on Fig.3A in the Manuscript?

Response 1-1: As the reviewer pointed out, we agree the difficulty in interpreting these results and toned down our statement in the revised manuscript (page 7, lines 266 through 268). In Fig. 3A and 3B, we separately prepared two membranes for GFP and pMYPT1 antibodies, not sequential re-probing. We failed to trim the area and missed 1 lane for anti-GFP blot (left panel) in Fig. 3A original image, which was corrected in the revised original image file. The left set was shown in Fig. 3A in the main text.

Point 1-2: Additionally, I would recommend here to repeat the analysis with some ‘standard’ loading control suitable and external for the proposed experimental setup or if already presented membranes are preserved and available to re-probe them with antibodies recognizing ‘neutral’ loading control;

Response 1-2: As the reviewer pointed out, some loading control would be desirable for these experiments. To our regret, these samples ran out and the reprobing of the presented membranes did not work very well as mentioned in Response 1-3. Under our experimental conditions, because constant numbers of cells were subjected to the analyses and the expression levels of ectopic GFP-MYPT1 were almost comparable, the total protein amount for each lane is thought to be almost equal. We believe that the pMYPT1/GFP-MYPT1 ratio is a reasonable indicator showing that the MYPT1 fragments inhibit the endogenous Rho-kinase activity in COS7 cells as the supporting evidence. We previously demonstrated the similar experimental design and data to these experiment (Amano et al. JCB 2015).

Point 1-3: To have better view on phosphorylation of endogenous MYPT1 I suggest to additionally use antibodies recognizing the total native MYPT1 and then compare the signals. It is still not ideal because of the influence of native MYPT1 from un-transfected cells but can give additional point of reference.

Response 1-3: According to the reviewer’s suggestion, the membranes previously blotted with GFP Ab in Fig. 3 were subjected to reprobing with MYPT1 Ab, which are shown in the revised original image file. Unfortunately, these membranes don’t seem to be evenly reblotted and suitable as figures in main text, probably because they have been left for a long time.

Point 1-4: Why membrane with GST antibodies do not align with the rest of membranes?

Response 1-4: We apologize that we deposited the incomplete version of original image file, and replaced by the revised version.

Point 1-5: The quantitative WB method should be precisely described.

Response 1-5: According to the reviewer’s comment, we described the quantitative immunoblot method in the section 2.4 of Materials & Methods (page 3, lines 130 through 140).

Point 2: MYPT1 mutants with deletions:

Point 2-1: As Authors have shown some of MYPT1 deletions disrupt its secondary structure and thus the question arise if the lack of phosphorylation in some of these mutants in not just the lack of interaction between the distorted mutant and Rho-kinase.

Response 2-1: As the reviewer pointed out, we cannot exclude the possibility and described it in the revised manuscript (page 15, line 523). We added the alanine-scanning data of DMs as Fig. S9 in the revised manuscript according to the reviewer’s comment (point 9), in which some charged amino acids appeared to be critical for the interaction with Rho-kinase. Even though the possibility of influence on the structure still remains, the importance of DM regions as binding interfaces was also shown by the milder amino acid substitutions than deletions (Fig. S9). We also demonstrated that PS1+DM3 and PS2+DM2 peptides, but not each PS or DM peptide, directly inhibited the Rho-kinase activity in vitro. Collectively, we believe that the DMs identified in our study increase the affinity to Rho-kinase and the resultant phosphorylation by Rho-kinase.  

Point 2-2: Additionally, I still encourage to use PDB files for both analyzed proteins but not to show the interaction of MYPT1 and Rho-kinase but to point on existing structures suggested regions of interactions as in proposed cartoon model. In my opinion using native structures could much better reflect the interaction potential.

Response 2-1: We thank the reviewer for the suggestion. According to the reviewer’s suggestion, we showed the locations of potential interfaces between MYPT1 and Rho-kinase on the already-reported structures (PDB code 2kjy and 2f2u) in Fig. S11 in the revised manuscript.

Minor:

Point 3: Line 25: please be precise and write exactly which DM of MYPT1 could serve as a potent inhibitor: all of DMs or just one or more and if more, which ones?

Response 3: According to the reviewer’s comment, we mentioned the combinations of PS and DM showing inhibitory effects in the revised manuscript (page 1, lines 25 through 26).

Point 4: Unification of abbreviations should be applied: DMs or DM(s);

Response 4: We changed DM(s) to DMs in the revised manuscript.

Point 5: Genetic preparation of phosphomimetic variants could be described;

Response 5: We have never tested the phosphomimetic version of mutants and peptides. Generally speaking, because the affinity between enzyme and product is lower than that between enzyme and substrate, we speculate that the phosphomimetic mutants/peptides have similar properties to those of phosphorylation-defective mutants in this study.   

Point 6: It would be nice to explain ‘DN’ abbreviation from figure 3 in figure legend or in the main text;

Response 6: According to the reviewer’s comment, we explained “DN” abbreviations in figure legend (page 8, line 282) and in the main text (page 7, line 251).

Point 7: Some plots still miss their headings;

Response 7: According to the reviewer’s comment, we added the heading to Fig. 3A and B in the revised manuscript (page 8, lines 273 through 274).

Point 8: I believe it would be easier for Readers to follow the story about DMs with the information about DM3 placed in the text before the figure and Table with it;

Response 8: According to the reviewer’s suggestion, the paragraph describing DM3 was moved to the front of Fig. 4 in the revised manuscript (page 9, lines 335 through 343).  

Point 9: I encourage to show all of so called ‘unpublished data’ (line 1059)

Response 9: According to the reviewer’s suggestion, we added the alanine-scanning data of DMs in Fig. S9 in the revised manuscript.

Point 10: Lines: 1085-1099 are chaotically written and hard to understand. I suggest to re-write this fragment and underline its sense justifying its presence.

Response 10: According to the reviewer’s suggestion, we modified this paragraph in the revised manuscript (page 15, lines 527 through 534).

Reviewer 4 Report

Now the authors have revised according to reviewers' comments and suggestions. Now it's ready for publication after minor text/language editing.

Author Response

We sincerely thank the reviewer for the suggestion and made sure thoroughly the manuscript. We are pleased to know that our revised manuscript is now acceptable for publication in Biomolecules.